# Review on Hybrid Reinforced Polymer Matrix Composites with Nanocellulose, Nanomaterials, and Other Fibers

**DOI:** 10.3390/polym15040984

**Published:** 2023-02-16

**Authors:** Mehmet Özgür Seydibeyoğlu, Alperen Dogru, Jinwu Wang, Mitch Rencheck, Yousoo Han, Lu Wang, Elif Alyamaç Seydibeyoğlu, Xianhui Zhao, Kimberly Ong, Jo Anne Shatkin, Siamak Shams Es-haghi, Sunil Bhandari, Soydan Ozcan, Douglas J. Gardner

**Affiliations:** 1Advanced Structures and Composites Center, University of Maine, Orono, ME 04469, USA; 2Department of Materials Science and Engineering, Izmir Katip Celebi University, Izmir 35620, Turkey; 3Aviation HVS, Ege University, Izmir 35040, Turkey; 4Forest Products Lab, Madison, WI 53704, USA; 5Manufacturing Sciences Division, Oak Ridge National Laboratory, Oak Ridge, TN 37381, USA; 6School of Forest Resources, University of Maine, Orono, ME 04469, USA; 7Department of Petroleum and Natural Gas Engineering, Izmir Katip Celebi University, Izmir 35620, Turkey; 8Vireo Advisors, LLC, Boston, MA 02130, USA; 9Department of Chemical and Biomedical Engineering, University of Maine, Orono, ME 04469, USA

**Keywords:** natural fibers, hybrid, nanocomposites

## Abstract

The use of composite materials has seen many new innovations for a large variety of applications. The area of reinforcement in composites is also rapidly evolving with many new discoveries, including the use of hybrid fibers, sustainable materials, and nanocellulose. In this review, studies on hybrid fiber reinforcement, the use of nanocellulose, the use of nanocellulose in hybrid forms, the use of nanocellulose with other nanomaterials, the applications of these materials, and finally, the challenges and opportunities (including safety issues) of their use are thoroughly discussed. This review will point out new prospects for the composite materials world, enabling the use of nano- and micron-sized materials together and creating value-added products at the industrial scale. Furthermore, the use of hybrid structures consisting of two different nano-materials creates many novel solutions for applications in electronics and sensors.

## 1. Introduction to Hybrid Reinforced Polymer Composites

Composite materials date back to approximately the first years of world evolution (bones, wood, etc.) with wheat fiber-filled bricks made from clay, soil, and sand [1]. Combining materials to optimize material properties has always been a promising approach for producing novel materials. In the early 1920s, steel-reinforced concrete changed the construction industry [2,3].

Although glass fiber (GF) is still the predominant fiber used industrially, many different fibers have been used for various engineering challenges. Among the various fibers, carbon fibers (CFs) were a choice material of the late twentieth century, used to replace many metallic parts with CF-reinforced components, thus affecting aerospace and many other industries [4,5]. Besides CFs, aramid fibers, with high thermal stability, have been used in numerous applications, including ballistics, high-pressure automotive hoses, and high-temperature materials. Basalt fibers are also garnering attention for civil engineering applications and special use in nuclear plant construction [3]. A new generation of high-performance fibers are ultrahigh-molecular weight polyethylene fibers, which have diverse applications [5]. Ceramic and boron fibers have also been used for highly specific and limited uses in high-temperature applications [6].

Among the various available resources, environmentally-friendly and sustainable fibers and natural fiber-reinforced composites have been an important alternative to conventional fiber-reinforced composites [7,8]. Natural fibers are primarily comprised of cellulose, hemicellulose, and lignin [9]. However, some natural fibers (wool, hair, silk, etc.) are comprised of protein. The percentages of these components differ from plant to plant. Among the natural fibers, wood flour has been used commercially for more than 100 years, and new products use bagasse, flax, and hemp fibers. Many natural fibers are being studied extensively [10].

With the introduction of nanotechnology enabled by advanced microscopy techniques, various nanomaterials have been discovered and used in polymer composites and other composites. Among the many nanomaterials, nanocellulose has been one of the most important materials, presented as “the future of materials,” and has been examined in numerous studies over the past two decades [11,12,13,14].

Composite materials have been formed using various fibers, as listed, as well as many different resins, including thermosets, thermoplastics, and elastomers [15,16]. In the search for new composites, researchers have found new ways to optimize the final materials properties using hybrid material techniques [17]. In composite materials, researchers begun using hybrid fibers for the reinforcement of polymers, and many studies have examined hybridizing nanomaterials, as well. Many studies have compared different fibers in the same matrix; however, in the past decade, the use of hybrid fibers and materials has created many novel results and products for various applications [1].

The hybridization of reinforcing phases is one of the most important topics for composite materials in terms of precisely optimizing the properties of the materials. Composite materials can have the following hybrid forms [18]:Hybrid composite system containing at least two reinforcing fibers;Hybrid composite system containing fibers and micron-nanoscale particles;Hybrid composite system containing at least two nanomaterials.

In this review, hybridization with at least two reinforcing fibers is described briefly. The use of hybrid fibers creates much better properties when the fibers are used solely in polymer matrices. This review highlights all of these improved properties. In Section 2, nanocellulose-reinforced composites are discussed. In Section 3, composites containing nanocellulose- and fiber-reinforced hybrid systems are discussed [19]. In Section 4, two nano-systems are explained in detail. Section 5 discusses the applications of these novel materials. In Section 6, concluding remarks are presented.

### 1.1. An Overview of Hybrid Composites

Hybrid composites are among the multifunctional materials used in advanced structural components when more than one characteristic benefit, such as mechanical strength and/or conductivity, is needed from the materials they contain. Hybrid composites are materials in which one type of reinforcing material is incorporated into two polymer matrix mixtures, or a particular polymer matrix is reinforced with more than one reinforcing material [20].

Such hybrid forms affect the weight and geometry of the advanced structural components used in specific applications [21,22]. Natural fibers are classified as being from flora, fauna, and minerals. Animal fiber components are typically based on proteins. Plant fibers (PFs) primarily comprise stem, leaf, root, wood, and straw which contain lignocellulose. Artificial fibers, however, are materials with synthetic chemical content and are obtained from fossil sources such as petroleum or coal [3]. Figure 1 shows the classification of fibers with examples [23].

The orientation of the fibers used for reinforcement in composite materials and the manufacturing processes used significantly affect the resulting mechanical properties. Therefore, the fiber orientation and textile processes used are critical when designing hybrid composites. Polymers are widely used as matrix materials in hybrid composites [24]. Discussions on matrices for composites are beyond the scope of this review.

The strength of hybrid composites depends on the properties of the fiber, the fiber length and diameter, fiber orientation, fiber aspect ratio, homogeneous distribution of fibers, matrix/fiber interface bonding, layers, and fiber textile construction type [20,25,26]. Deciding on the type and properties of fibers or fillers is of primary importance in hybrid composite design and production. Hybrid composites can be designed according to the number of layers and different forms and orientations of the reinforcing fibers, as shown in Figure 2 [27]. Figure 2 shows different hybrid structures with distinct morphologies.

A subset of hybrid composites is defined as hybrid biocomposites when one of the elements consists of lignocellulosic fibers. While a natural biofiber is used as reinforcement material in these composites, a non-biodegradable polymer or biopolymers are used as the matrix material. Some hybrid biocomposites also contain synthetic fibers as well as hybrid fibers. Hybrid biocomposites are either completely environmentally-friendly hybrid biocomposites or partially environmentally friendly [28,29]. The completely environmentally-friendly ones are biopolymer matrix structures reinforced with or without cellulosic fibers (i.e., green biocomposites). Hybrid biocomposites are affected by fiber/matrix interfacial bonds and the characteristics of their sustainable components. Natural cellulosic PFs can be used in sizes ranging from micrometers to centimeters.

The mechanical performances of hybrid biocomposites examined in previous studies are shown in Table 1 [30]. Most of the studies in the literature focus primarily on mechanical properties, the improvement of interfacial bonding, and fabrication methods. The parameters investigated in hybrid composites are the chemical modification of fibers, fiber surface treatments, crystallinity, weathering resistance, strength, and thermal stability [31,32].

Natural fibers which are cellulose-based are hydrophilic because they contain hydroxyl groups on the surface and in the bulk. Therefore, they swell when they interact with water or moisture, resulting in unsatisfactory properties. To eliminate this problem, chemical modification of the fiber can be carried out. However, most of these methods are carried out with chemicals that are harmful to the environment [33]. An alternative to chemical processing is hybridization, in which natural fibers are combined with synthetic hydrophobic fibers. This method allows for a simultaneous reduction in moisture absorption and improvement in the properties of the designed hybrid composite.

In addition to synthetic fibers, natural fibers are reinforced to improve the properties of the materials designed and produced in hybrid composites, reduce moisture absorption, balance the costs of the fibers, and reduce the negative environmental impact, and energy and carbon footprint [34].

Nanocomposites are possible alternative materials to overcome the disadvantages and inadequacies of traditional composites. Nanocomposites are composites in which nano-sized fillers (nanoparticles) are added to reinforce the polymer matrix to improve specific properties of the material. With the addition of nano-sized fillers to the polymer matrix, changes in chemical, electrical, mechanical, and physical properties are observed. Researchers in applied materials science have conducted intensive studies on nanocomposites [6].

By adding nanoparticles to the matrix material, the material’s mechanical behavior can be changed without changing the total mass of the structure. Hybrid nanocomposites are obtained by adding nano-sized fillers in addition to fiber-reinforced or nanoparticle-doped polymer matrix composites. When a biodegradable product is used as a supplement, the term biocomposite can be used. For example, hybrid nanobiocomposites are formed by the incorporation of nano-sized filler into polymer matrix composites reinforced with natural fibers.

The surface areas of nano-sized materials are quite high [35]. Therefore, the inclusion of such substances in the structure affects the connection between the fiber and the matrix, and enables the improvement of mechanical properties [23]. Including particulate reinforcements in composite structures increases stiffness, reduces cost, and prevents discontinuities. It provides a more efficient load transfer between the fibers and the matrix [36].

### 1.2. Manufacturing Methods

Composite parts are produced through many different methods. For the successful production of a material or a component, the method must be cost-effective and reliable [37,38]. The cost-effective paradigm is highly dependent on production speed, consumables, and infrastructure requirements. For reliability, all parts after production must be of the same quality. The part must be able to be shaped into the desired geometry, be in suitable tolerances during shaping, and show the expected mechanical properties which were determined during the design. In line with these requirements, different manufacturing methods have been used.

Each method provides inherently different advantages. The selected matrix materials and reinforcement elements are important to the production method of hybrid composites. It is critical to ensure a homogeneous distribution of fibers and/or particles in hybrid composites containing at least two reinforcing fibers, containing fibers and micro-nano-scale particles, and containing at least two different nanomaterials.

Hybrid composites, especially with thermoset matrices, can be produced by a wide variety of methods, such as autoclave molding, cold press, compression molding, hand lay-up, hydraulic pressing, vacuum bagging, and infusion methods [20,39,40,41]. Extrusion, injection molding, and thermoforming are widely used as production techniques in thermoplastic composites [42,43]. Other production methods include spray-up, filament winding, pultrusion, and additive manufacturing [44,45].

Processing methods for hybrid composites are listed in Table 2 [15,16].

### 1.3. Applications

Various nano-sized fillers are used to enhance the properties of polymer matrix composites. Shanker and Rhim reported that nanofillers can be used in the formation of hybrid nanobiocomposites [46]. The most suitable and promising carbon nanofillers, carbon nanotubes (CNTs), carbon nanofibers (CNFs), graphite, graphene, and nano-clay are used to develop hybrid nanobiocomposites with multifunctional applications. Among these, graphene and CNTs are the most widely-used nanofiller materials because of their superior morphological properties, structures, and physicochemical properties that can have applications in high-strength composites.

The addition of different nano-sized materials (1–5 wt. % of nanoparticles, fly ash, and metallic nanoparticles) to hybrid nanobiocomposites reinforced with natural fibers increases tensile strength and impact strength compared with epoxy matrix materials reinforced with only one natural fiber type [47,48,49]. Furthermore, significant changes in material behavior are occurring and can lead to new applications in composite-based industries. More studies on this topic will enable the usage area to become widespread.

Further research on biofiber/nanofiber-based hybrid composites is crucial for automotive parts, ballistics, and biomedical applications. The increase in fuel costs, the desire to increase the useful load-carrying capabilities of aircraft, and the desire to improve their maneuverability has motivated scientists to research innovative materials with low weight and excellent mechanical properties [50]. Puttegowa et al. showed the important requirements of aerospace and defense industry components and the effects of these requirements on structural design [30]. In the aerospace industry, many commercial aircraft manufacturers have turned to the use of low-cost hybrid composites with customized features to meet the high demands regarding energy and safety. In addition, hybrid composites have greater elastic strain energy, storage ability, and strength, and a higher strength-to-weight ratio than conventional metallic materials such as aluminum and steel. The use of composite materials in the production of aircraft structural components has been supported by the development of strong reinforcing materials such as glass, CFs, and advanced polymers [51].

The development of strong reinforcing materials such as glass, CFs, and advances in polymers have also been supported in the development of suitable materials that can be used in the manufacture of the latest aircraft components [51]. In the latest technological studies, glass and CFs such as artificial fibers and flax, kenaf, and jute have been used as biofibers for hybrid composites reinforced with biofibers.

### 1.4. Nanocellulose

Figure 3 shows the classification of cellulose nanomaterials (CNMs) by ISO/TS 20,477 and the modified classification used in this review. Each subclass (subcategory) is indented farther than its superordinate class; sibling classes are listed with the same indentation. For some classes, a few subclass terms are also listed. The class and subclass terms are not comprehensive; additional classes and subclasses can be added. For example, cellulosic materials do not only include CNMs and microcellulose. The terms in Table 1 and Figure 1 are defined by ISO/TS 20,477 and ISO/TS 80004, or are explained by IUPAC [52].

In the modified classification, for simplicity, the term *nanocellulose* is substituted for *cellulose nano-object*. *Nanomaterial* refers to a substance or material that is in the nanoscale and at least in one dimension less than 100 nm, and *nanocellulose* is a parallel term. Although nanomaterials include those having internal or surface structures according to ISO/TS 20477, nanocellulose generally includes those with any external dimension in the nanoscale, equivalent to nano-object, including cellulose nanofibers, nanofilms, and cellulose nanoparticles. ISO/TS 8004-2 defines *nanoparticles* as materials with all external dimensions in the nanoscale; the lengths of the longest and the shortest axes do not differ significantly. In most of the literature, *nanoparticle* is inclusive of various-shaped nano-objects. By this definition, there are rare reports of cellulose nanoparticles in the literature. According to ISO/TS 20477, *nanofiber* is a nano-object with a significantly large aspect ratio. It should include cellulose nanocrystals (CNCs) and cellulose nanofibrils; however, this understanding is not in agreement with ISO/TS 20477, which states that *nanofibril* and *nanofilament* are equivalent terms to *nanofiber*. CNCs obtained from wood or other cellulose sources have been reported to have diameters and lengths in the range of 3–5 nm and 100–300 nm, meaning that their dimensions meet both the definition of nanofiber according to both ISO/TS 20,477 and 80004-2, implying that they are both a nanocrystal and a nanofiber. Furthermore, if electrospun cellulose fibers have diameters in the nanoscale, they should also be classified into nanofibers [53], although their crystalline structure is cellulose II. Assigning the terms *nanolayer*, *nanocoating*, and *nanofilm* to *nanocellulose* (i.e., nano-objects) and the terms *nanostructured films*, *coatings*, and *layers* to *nanostructured material* follows the hierarchy of nanomaterial terms in ISO/TS 80004-11.

Similar reasoning follows for the term micro-cellulose, after nanocellulose, referring to any cellulose materials in the microscale, such as those having one or more dimensions less than 10 μm. When the diameter of cellulose fibers is less than 10 μm, the fibers can be classified as cellulose microfibers. Cellulose microfibers are often manufactured cellulose fibers with diameters in the range of 100 nm to 10 µm, which includes most electrospun cellulose fibers [53]. Most PFs are larger than 10 µm and, hence, cannot be classified as microfibers. In general, when naming an object, the name of the material is not typically the head noun but, rather, the modifier. However, naming micro- or nano-fibrillated cellulose does not follow this standard. The term micro-fibrillated cellulose emphasizes the fibrillation process that produces it by using a size prefix with a process. Micro-fibrillated cellulose could have been better named cellulose microfibrils;however, the term microfibril has been historically used by plant anatomists to designate the cellulose organization structure in plant cell walls along with microfibril and elementary fibril. Isolated cellulose in the micrometer range should not use the term microfibril to avoid confusion. With this consideration, use of the term micro-fibrillated cellulose should be discouraged.

Microcrystalline cellulose consists of fragmented cellulose fibers at a size distribution around tens of micrometers with small aspect ratios. It could be better named cellulose microparticle, cellulose fines, or cellulose microcrystal, as suggested by ISO/TS 20477. However, the term has been widely accepted commercially and in the pharmaceutical industry as an excipient. The modifier microcrystalline or nanocrystalline designates the state of material with crystallites in the micrometer or nanometer range and does not give clues about external dimensions. Nanocrystalline materials generally refer to polycrystalline materials with grains in the nanoscale or semicrystalline materials with a significant fraction of dispersed nanoscale crystallites in a matrix of the amorphous phase. Therefore, charcoal treated at high temperatures beyond 1400 °C can also be classified as a nanocrystalline material since it contains nanoscale carbon crystallites in an amorphous carbon matrix (c). Nanocrystalline materials such as nanocrystalline metals and nanocrystalline ceramics often refer to bulk materials with a nanocrystalline phase. Following these practices, use of the term nanocrystalline cellulose to refer to CNCs should be discouraged.

The word fiber has many definitions. It can be a mass word referring to any object with an acicular shape with macromolecular chains usually orienting along the long axle. Those without oriented macromolecular chains might be called nanowire or nano-thread. In the materials, paper, and textile industries, fiber is a broad term referring to a type of cell or cell aggregate in hardwood, bast, and cotton fibers, as well as various polymeric, carbonaceous, and inorganic materials for textiles or reinforcements. The term cellulose fiber is also an inclusive concept referring to manufactured cellulose fibers such as rayon and lyocell fibers in the form of filaments, chopped short fibers (in most cases), chopped yarns, yarns, and fabrics. Cellulose ester fibers are also included in cellulose fibers. The core structure of cellulose fibers is the presence of linear chains of thousands of glucose units linked together, which enables significant hydrogen bonding between OH groups on adjacent chains, causing them to pack closely into fibrils. When fibers have diameters in the nanometer range, they can be classified as nanofibers. Since fiber is an inclusive term, nanofiber should also be inclusive of all fiber in the nanometer range. Fibers are usually presented in the form of yarns, fabrics, filaments, and chopped short fibers in filaments or yarns, as shown in Figure 4a.

Cellulose is sometimes used with carbon. There are many carbon nano-objects, i.e., nanocarbons, such as CNTs and graphene, which are classified as shown in Figure 2b. Researchers and industrialists have integrated the remarkable properties of individual nanocarbons into useful, macroscopic ensembles (i.e., carbon-nanostructured materials such as fibers, films, membranes, foams, and nanocomposites).

## 2. Nanocellulose-Fiber Hybrid Composites

### 2.1. Nanocellulose-Hybrid Fibers with Synthetic Fibers

The hybridization of multiple reinforcing additives has been heavily researched to synergize the benefits from each additive. The most researched reinforcing additives are GFs, CFs, and various natural fibers, such as wood fibers and PFs. The use of reinforcing fibers typically aims to improve the mechanical properties, although it is also used as a reinforcement for gas barriers. The mechanical performance of the composites hybridized with multiple additives, however, is slightly degraded because of the lack of performance of the natural additive [54]. Despite the slightly decreased performance of the hybridized composites, the benefits of the natural fibers are significant in terms of material sustainability and decreased carbon footprint.

Epoxy composites have been reinforced using mixtures of GF and sisal/jute fibers [55]. The fibers were laid to generate multiple layers in the dry state, and epoxy resin was added to the multi-layers to wet the fibers and was later cured. The composite materials showed excellent mechanical properties, although they did not show the interfacial interactions among the fibers, since the epoxy coated the fibers before the fibers were attached to others.

Epoxy, a thermosetting resin, is preferred for manufacturing hybrid composite materials, especially for application in the structures of aircraft, which requires low-dielectric constants that can be provided from the natural fibers [54]. The hybridizing of GF and kenaf fibers with epoxy resins showed good low-velocity impact performance, which is required in aircraft applications [54].

In general, the synergetic effects from the hybridized composite materials result in better performance than those predicted by the rule of hybrid mixtures [56]. In one study, a mixture of GF and CF was thermally compounded with polypropylene (PP) [55]. The strength of the hybridized composite samples showed a positive deviation from the elastic modulus and showed no existence of a hybrid effect. The hybrid reinforcement effect in mechanical performance was hypothesized by suggesting that microcracks starting at the ends of CFs do not lead to the composite’s failure because the GFs act as crack arrestors [56].

The hybrid fiber reinforcement was also demonstrated in another method for the orientation of additive fibers. A hybrid of CFs and GFs was thermally compounded with PP, and the compound samples were molded into tensile specimens by injection molding [57]. The authors reported that the fiber orientation was different in different layers, thus affecting the fracture toughness performance of the composite materials (Figure 5). Fiber orientation occurred during injection molding, depending on the species of the reinforcing fibers and flow direction of the composite melts. A more oblique orientation of the fibers contributed to the good mechanical performance of the hybrid composite materials. The hybridizing of two fibers tended toward parallel alignment to the molding flow direction.

The hybridized reinforcing additives can result in modification to the crystallization mechanisms of semi-crystalline thermoplastics. A mixture of nanoparticles of silica fibers and GFs was thermally compounded with PP with the aim of modifying the mechanical and crystallinity properties of the hybrid composite materials [58]. The PP matrix showed different crystallinities in the addition of hybrid additives, leading to a heterogeneous nucleating effect of GFs being larger than the effect of nanosilica with different nucleating mechanisms as shown in Reference [58]. The synergistic hybrid effects were noticeable in the reduced thermal degradation. Notably, the hybrid effects come from the additives’ mechanical properties and also from the different morphological properties of the additives.

Mechanical properties can be significantly improved by hybridizing the conventional reinforcing fibers and nanofillers of alumina, silica, clay, CNTs, and others [59]. The addition of a small amount of a nanofiller can improve the strength and elastic modulus of a composite. Alumina, silica, and CNT have positive effects on the strength of the composite, whereas clay decreases the mechanical properties.

The interfacial properties of hybrid composite materials are critical to their mechanical and physical performance. The type of surface treatments can be selected based on the target matrices and the reinforcing fibers. A study was conducted with various surface treatments of the reinforcing fibers of sisal and glass [60]. Chemical surface modifications, such as alkali, acetic anhydride, stearic acid, permanganate, maleic anhydride, silane, and peroxides, of the fibers and matrix were successful in improving interfacial adhesion and compatibility between the fiber and matrix. The nature and extent of chemical modifications were analyzed by infrared spectroscopy, and the improvement in fiber–matrix adhesion was determined by studying the fractography of composite samples via scanning electron microscopy.

A study was conducted to compare the reinforcing effects from CF and GF [61], which are two dominating reinforcements for polymer composites. Fiber-reinforced thermoplastics are widely used in many applications. The study reported that the mechanical performances of CFs/GFs were well matched to the result found by multiple sources in the literature, which is that CFs are 20% stronger in tensile and flexural strengths than GFs, but that GFs increased the elastic modulus to double that of CFs. Also, GFs can improve the impact resistance of the composite materials more than CFs.

Aramid fiber is a well-known reinforcing fiber, especially used for military applications because of its superior mechanical properties. Long fibers of aramid and kenaf were used in their woven forms, which is typical in military applications, to manufacture composite samples for ballistic testing [62]. The testing results showed that hybrid composites (14 layers of Aramid and 2 layers of kenaf) significantly improved ballistic performance as compared with other hybrid composites. The combination of natural fibers and aramid is rare since the aramid fibers are expensive and specialized for high-end applications, whereas natural fibers are more common in low-cost building material applications.

Over the past decade, researchers have examined the benefits of including nanocellulose in GF/CF composites for enhanced properties and performance. In a study comparing GFs and microfibrillated cellulose fibers in a polamide-6/PP blend, the tensile and flexural stiffness increased from the neat polymer blend [63]. Through incorporating GFs with microfibrillated cellulose fibers, the water absorption in the composite was decreased, which helps to improve the mechanical properties and ease of processability. The GF/microfibrillated cellulose fiber content was optimized at 15 wt. % and 10 wt. %, respectively; however, it could not match the mechanical properties of a 30 wt. % GF/polamide-6/PP composite.

One main area of nanocellulose use with GF composites is the coating of GFs with nanocellulose to enhance their mechanical properties. A research group found that, by immersing chopped GFs in an aqueous suspension containing 1 wt. % CNC, the interfacial shear strength (IFSS) of the fibers in an epoxy matrix was enhanced [64]. The proposed mechanism for IFSS enhancement is that the CNCs have a better chemical affinity to the epoxy than the GFs, thus increasing the interfacial adhesion between the fiber and the matrix. However, by coating GFs in a higher weight-percentage CNC solution, IFSS was found to decrease because of the inherent brittleness of the CNCs [64,65].

Traditionally, GFs are coated with CNCs through dip coating from an aqueous suspension. Some researchers have investigated alternative coating methods, such as CNC emulsions, spray coating, and slot die coating, all of which improve the IFSS compared with non-coated GFs [64,65,66]. The differentiator between these methods is the scalability; aqueous suspensions are more easily scaled than emulsions and spray coatings, and slot die coating methods are more easily scalable than dip coating. Regardless, using CNCs as a coating on GFs in epoxies has been shown to increase the IFSS when compared with GFs without CNC coatings.

GF/nanocellulose composites have also been investigated for sheet molding compounding (SMC) of epoxy composites [64,67]. By adding CNCs during SMC processing of GF/epoxy composites, the viscosity, flexural strength and stiffness, and tensile strength and stiffness increased. The major advantage of using CNCs in SMC is the ability to achieve similar properties while reducing the overall weight of the composite. Asadi et al. replaced 10 wt. % GFs with 1.5 wt. % CNCs in SMC, producing a lighter-weight composite while still achieving enhanced mechanical properties [67]. Overall, the use of nanocellulose with GF composites has mainly focused on using CNCs as a GF coating in epoxy matrices, although it has been used in other ways to create lighter-weight composites.

Shariatnia et al. coated CFs with CNCs and CNTs in CF-reinforced epoxy matrix samples. They used CNCs to disperse and stabilize the CNTs. In hybrid CFRPs, the addition of CNCs and CNTs increased flexural stiffness by 15% and strength by 33% [68]. According to Kaynan et al., CNCs were used to effectively coat CFs with CNTs and graphene nanoplatelets (GnPs). CNCs provided a suitable platform for designing the interface of the hybrid composites [69]. These studies showed that CNCs are a good dispersing aide for nanomaterials and are easily coated onto CFs. Shao et al. studied the fatigue performance of CNF-doped, CF-reinforced epoxy composites. An increase in adhesive strength between CF and the epoxy matrix was observed, which is attributable to physical modification with CNFs [70]. Chai et al. investigated the damping properties of CF epoxy hybrid composites reinforced with polysulfone/CNC nanofiber membranes using dynamic mechanical analysis tests. In this study, the 10% polysulfone/0.5% CNC membrane additive increased the damping properties of CF epoxy hybrid composites with minimal loss of mechanical performance [71].

The major motivation of research in hybrid composites is to maximize the effects of reinforcements from various additives, leading to synergistic property improvements that are greater than singular reinforcing additives. The hybrid effects have been proven in many polymeric matrices; however, to be certain of the synergistic effects, the most important aspect of hybrid composite manufacturing is how to design the hybrids according to the target properties. Some of the most sought-after properties include mechanical, thermal, gas permeability, optical, and material sustainability with the aim of low carbon footprints and environmental protection. A comprehensive understanding for each additive is required to obtain the desired hybrids designed for specific material property targets. In some cases, hybridization can be designed to adjust the strength performance of the composite materials by partially replacing the expensive additives with less expensive but weaker additives, thus creating a balance between performance and manufacturing costs.

### 2.2. Nanocellulose-Hybrid Fibers with Natural Fibers

Natural fiber–reinforced polymer (NFRP) composites are a unique member of the fiber-reinforced polymer (FRP) composite category. Compared with FRPs that contain synthetic fibers (GFs, CFs, etc.), NFRPs provide balanced benefits among mechanical performance, cost, and environmental impact [72]. However, for more performance-demanding applications, NFRPs are often mixed with high-performance (nano) fibers to improve performance [73]. Nanocellulose has been investigated as a replacement for traditional nanomaterials to reinforce polymers because of their superior (estimated) mechanical properties [74].

Early studies of NFRPs started with thermoplastics as the polymer matrix. CNFs were used by Li et al.to reinforce wood fiber-filled HDPE composites [75]. While keeping the weight percentage of HDPE constantly at 40 wt. % and the coupling agent (maleic anhydride grafted polyethylene) constantly at 3 wt. %, the wood flour content was changed as the CNF content was changed from 5 wt. % to 20 wt. %. Powdery polymers were blended in CNF suspension and freeze dried before extrusion compounding and compression molding. The mechanical properties of wood-filled HDPE composites were largely increased as CNF content increased. For example, the bending strength, stiffness, impact strength, and storage modulus of wood–plastic composites (WPCs) were improved by approximately 50%, 30%, 60%, and 20%, respectively, with a 10% CNF loading level. However, no data were reported for the tensile properties of the composites. In another study, only CNF suspension and wood flour were premixed, dried (oven drying or freeze drying), and milled before being compounded with PP and maleic anhydride-grafted PP [76]. Oven drying caused more severe agglomeration of CNFs than freeze drying, as shown in Figure 8. A fiber content of 3 wt. % freeze-dried CNFs was identified as the best reinforcement for wood flour-filled PP composites; the bending strength, bending modulus, tensile strength, and impact strength of the WPCs were improved by 5%, 10%, 7%, and 9%, respectively. Dried CNCs (1 wt. %) and maple wood flours (20 wt. %) were directly melt compounded with PP and maleic anhydride-grafted PP (2 wt. %) before compression molding [77]. The mechanical properties of CNC-reinforced WPCs, in most cases, were not significantly different from the original WPCs, likely because of the lower CNC content and the agglomeration of CNCs during drying. Apart from the powder form, a preform of bamboo fibers and CNFs was prepared by compression molding the dehydrated mixture of those fibers after they were comingled in an aqueous suspension [78]. The preforms were then compression molded with polybutylene succinate (PBS) to form hybrid composites. Although the hybrid composites exhibited higher tensile properties than pure PBS, no experiments were carried out for bamboo fiber-filled PBS composites, making it impossible to analyze the individual effects of CNFs.

Besides thermoplastics, natural nanofibers have also been used to reinforce WPCs with thermoset matrices. Oil palm empty fruit bunch nanofibers (3 wt. %) were incorporated into epoxy resin and then infiltrated into non-woven kenaf mats to form hybrid composites [48,79,80]. In the kenaf/epoxy composites, the nanofibers improved the storage modulus by 20% and increased the glass transition temperature by 8 °C, which was attributed to high stiffness [48].

## 3. Nanocellulose/Nanomaterial Hybrid Polymer Composites

Nanocellulose materials have exceptional properties, as explained in previous sections. Carbon nanomaterials with outstanding properties in various forms have numerous applications. This section presents detailed explanations of hybrid structures of nanocellulose and other nanomaterials.

### 3.1. Nanocellulose-CNT Hybrids

The use of different materials in hybrids creates novel properties from the combined reinforcements. The use of nanocellulose and carbon nanomaterials builds environmentally-friendly, strong, nontoxic, and bio-based solutions for various carbon nanomaterial applications. Furthermore, nanocellulose generates dimensional stability without any requirement for a polymeric matrix [81,82]. Carbon-based materials are a large category of materials that includes active carbon, coke, CFs, and many advanced carbon nanomaterials [4]. Since the discovery of the carbon nanomaterial fullerene in 1988, there has been tremendous research on carbon nanomaterials from various disciplines, including physics, chemistry, materials science, and several engineering fields, such as electronics, biomedicine, chemical engineering, and mechanical engineering [83,84,85,86,87].

Since the first report on CNTs published by Sumio Iijima in 1991 [88], significant efforts have been made to strengthen materials (especially polymer-based materials) with nanotubes [89]. Since their discovery, CNTs have attracted considerable attention because of their unique properties, such as high electrical conductivity; good chemical, mechanical, and thermal stability; high tensile strength; and ductility [90]. Synthetic polymers are the most-used class of matrix materials for nanotubes. The production of CNT/polymer composites requires the dispersion of nanotubes in a polymer matrix. Generally, dispersion is required to separate CNT bundles into stable few- or single-CNT colloid particles. In addition to the difficulty of homogeneous distribution in the matrix, the production issues of nanostructured materials increase the cost of nanocomposites [91]. Therefore, these structures are preferred as a secondary reinforcement and are a subject of interest in hybrid composite studies [92].

Nanocellulose/nanocarbon composites are a class of hybrid composites that contain cellulose and carbon nanoparticles [82]. Nanocarbon materials are typically combined with nano-cellulose to create environmentally-friendly, low-cost, durable, dimensionally-stable, non-melting, and non-toxic materials. In addition to its advantageous combination with nanocarbon materials, nanocellulose is an attractive material for biomedical applications, which is attributable to its tunable chemical structure, non-animal nature, and viscoelastic properties [93,94]. Cellulose nanomaterials include cellulose nanofibers and CNCs [95]. The fibrillation of cellulose material can be accomplished using mechanical, chemical, enzyme methods, or combinations thereof. After fibrillation, the width of cellulose nanofibers is typically between 3 and 100 nm, and the length can be several micrometers [96]. To obtain CNCs, the crystalline portions should be separated from the amorphous regions of the fibers or fibrils via acid hydrolysis [97]. Cellulose nanofibers are longer in length, whereas CNCs have a similar diameter, a shorter needle or rod-like morphology, and a harder structure.

The properties of nanocellulose materials are often variable, depending on their origin, type, and processing method. These factors all affect the properties of a composite or hybrid structure and are obtained from its combination with other materials. Nanocellulose/nanocarbon hybrid composites exhibit more advantageous properties than composite materials containing only cellulose nanoparticles or only carbon nanoparticles. Nanocarbon reinforcement can further increase the mechanical strength of nanocellulose materials. In addition, nanocellulose materials also support the flexibility of nanocellulose/nanocarbon hybrid composites. Typically, hybrid composites are produced from aqueous dispersions of nanocellulose and carbon nanoparticles [98,99]. Cellulose nanoparticles prevent the aggregation of carbon nanoparticles and ensure that they are kept in stable homogeneous suspensions. Furthermore, they are excellent dispersing agents for carbon nanoparticles [100,101]. Other advantageous properties of nanocellulose/nanocarbon composites are their thermal stability, tunable thermal conductivity, and optical transparency [102,103]. Nanocellulose/nanocarbon composites can be used in a wide variety of industrial and technological applications, such as water purification, isolation, the separation of various molecules, energy generation, storage and conversion, biocatalysis, food packaging, and the construction of fire retardants [103]. The hybrid composites are also used as fillers for various materials, often polymers, to improve their mechanical, electrical, and other physical and chemical properties [102,103,104,105].

Among carbon nanomaterials, CNTs have been studied since 1991, and graphene was discovered in 2004. Other forms of carbon nanomaterials include CNFs, nano-diamonds, fullerenes, and quantum dots; however, these four groups are out of the scope of this review. Carbon is a common element in nature and the most capable element that can form compounds with many elements in the periodic table [89]. As a unique chemical element, carbon can form different types of bonds and bond several atoms [106,107]. In the past two decades, innovative forms of carbon have emerged that have a nanostructure. Carbon nanomaterials contain a rich polymorphism consisting of several allotropes of different dimensionality. Allotropy is the property of some chemical elements which allows them to exist in two or more different forms or allotropes when found in nature. There are several allotropes of carbon: fullerene molecules (0D), nanotubes (1D), graphene (2D), and nano-diamonds (3D) [108]. All allotropes exhibit different chemical and physical properties, and carbon nanostructures are very important for nanoscience and nanotechnology. In particular, carbon nanostructures are characterized by their excellent electrical conductivity, superior mechanical strength, exceptionally high surface area, high transparency, and structural stability [109,110].

CNTs are one of the most important nanomaterial components. CNTs have unusually high strength and stiffness and are also chemically stable and electrically conductive [91]. These properties make the use of CNTs attractive for the design of composite materials based on different matrices. Composites are used in many fields, including energy, aerospace, defense, automotive, marine, rail systems, construction, biomedicine, and electronics [89,106,111]. CNTs are among the strongest and most conductive nanomaterials known and are, therefore, of great interest for nanocomposite structures. Single-walled nanotubes can add conductivity and increase the mechanical strength of composite structures. With these benefits, they are used extensively in the production of hybrid composites [112]. A schematic overview of CNTs and CNFs is shown in Figure 6.

CNTs have a high length-to-diameter ratio and, thus, a relatively large surface area. Diameters can reach nanometers and lengths can reach several micrometers and even centimeters. Because of these properties, CNTs are suitable candidates for hydrogen storage, the removal of pollutants from water and air, and drug delivery. CNTs have excellent mechanical properties that are mainly attributable to sp2 bonds. CNTs are similar to CNFs in their morphological and mechanical properties. For example, highly crystalline, thick tunicate CNFs exhibit strength close to those of commercially available multi-walled CNTs (3–6 GPa) [113]. However, the strengths of other CNF types are lower; for example, the average strength of wood-derived CNFs ranges from 1.6 to 3 GPa [113]. Therefore, CNTs increase the mechanical strength of nanocellulose/CNT composites. Zhu et al. studied polydimethylsiloxane polymers for strain sensor applications by using nanocellulose and CNTs [114]. Zhan et al. studied polypyrolle hydrogels reinforced with CNTs and nanocellulose for supercapacitor applications [115]. As a result, nanocellulose/CNT composites are used in similar industrial applications as nanocellulose/graphene composites. They are used in power generation, energy storage and conversion, filling polymeric materials, sensors, and biosensors.

### 3.2. Nanocellulose and Graphene and 2D Nanomaterial Hybrids

Graphene is a 2D sheet of sp2-hybridized carbon atoms [108]. In this structure, the carbon atoms are bonded in a honeycomb form with strong bonds in one plane [111]. Graphene, which is an allotrope of carbon, is in a form in which carbon atoms make covalent bonds in one layer [116,117]. Graphene has the thickness of one atom and has various unique properties [118]. It is stronger than diamond, more conductive than copper, and more flexible than rubber, making it attractive for research and development [110]. The intrinsic mechanical properties of free-standing, single-layer graphene sheets have been measured based on nano-indentation or nano-line using atomic force microscopy. The Young’s modulus was measured to be 1 TPa, and the intrinsic tensile strength is 130 GPa. These properties make graphene the strongest material ever measured. Graphene has good thermal conductivity above 3000 W/mK. The thermal conductivity of single-layer suspended graphene at room temperature has been measured to be 3000–5000 W/mK. The optical absorption of graphene is πα ≈ 2.3%, and it can sustain extremely high densities of electric current [118,119,120,121]. A schematic overview of graphene and CNFs is shown in Figure 7.

Until the discovery of graphene, researchers widely believed that 2D crystals were not stable. Some experimental results suggested that this theory was correct until the discovery of graphene by Andre Geim and Kostya Novoselov in 2004 [122]. These scientists produced graphene using mechanically exfoliated graphite. The effects of different reinforcements in hybrid composite structures are critical [123]. In this context, graphene and graphene-based materials hold great promise not only for industrial and technological applications, but also for biomedical applications such as medicine, and gene and protein delivery; photothermal therapy; biosensor construction; bioimaging; and antimicrobial therapy. Furthermore, the addition of graphene makes materials electrically conductive. In the most-used nanocellulose/nanocarbon hybrid composites, graphene or CNTs are widely used as nanocarbons, and nanocellulose is used less frequently with nano-diamonds and, especially, fullerenes.

The combination of carbon nanomaterials and nanocellulose opens many new possibilities for new materials designed for multipurpose uses. These hybrid materials can be prepared in various 1D, 2D, and 3D forms [124,125]. The dispersion of carbon nanomaterials in certain fluids and polymers is one of the greatest challenges of nanotechnology [126], and significant research has been carried out to prepare long-term stable carbon nano-dispersions using chemical functionalization. The use of nanocellulose also helps to prevent agglomeration of the carbon nanomaterials without the need for surface treatments [100,101]. Similarly, as with fullerenes, cellulose nanoparticles in the form of nanofibrils and nanocrystals increase the dispersion of graphene nanoparticles in aqueous environments and prevent their aggregation without the need for chemical functionalization [100]. A water-based dispersion is the starting material for fabricating nanocellulose/graphene composites. These composites can be formed by filtration [127], filtration combined with hot pressing [128] to produce films, or freeze drying [129] and freeze casting [103] to produce 3D materials such as aerogels and foams.

The harmonious interaction of nanocellulose and carbon nanomaterials can be beneficial for different applications, including foams, sponges, and aerogels [101,130,131]. the use of carbon materials increases the strength, whereas the use of nanocellulose for carbon nanomaterials creates flexibility [132,133,134,135]. With the high electrical and thermal conductivity of carbon nanomaterials, hybrid materials with nanocellulose create bio-based sustainable materials with improved thermal conductivity and electrical conductivity [98,136,137]. The use of nanocellulose can provide optical transparency for carbon-based nanomaterials, and the use of carbon nanomaterials for nanocellulose can generate new applications in the field of photothermal properties [138] and has high adsorption properties [133,139]. These hybrid materials have many uses in the field of biomedicine [112]. In addition, 3D composite structure production and additively-manufactured (i.e., 3D printed) raw materials are produced using hybrid nanocellulose and carbon nanomaterials [134,140]. There are many studies on the electrospinning of cellulose and carbon nanoparticles, which can find many different uses [132,135,141]. Graphene increases the mechanical strength of nanocellulose/graphene composites and imparts electrical conductivity to them. Fan et al. studied the use of hybrid 2D graphene oxide and 1D nanocellulose material in waterborne polyurethanes for increased electrical conductivity and UV absorbance. They observed improved electrical conductivity for polyurethane materials [142]. Faradilla et al. studied the interaction of graphene and nanocellulose; the interface was improved using polyethylene glycol. They also observed that the lower molecular weight of polyethylene glycol was more effective for interfacial issues [143]. Wang et al. studied the interaction of nanocellulose and graphene oxide for capacitance properties; they improved the material properties using polypyrole, and these materials have a very good potential for energy storage applications [144]. Xu et al. studied the humidity sensing of polyvinyl alcohol (PVA) using nanocellulose and graphene oxide. Nanocellulose helped to disperse graphene in the PVA polymer [145].

To conclude, nanocellulose and carbon nanostructures offer many new alternatives for various polymer matrices, contributing to diverse applications.

### 3.3. Nanocellulose and Nanoclay Hybrids

The excellent properties of nanocellulose [14,146] make this material a good candidate for various and growing applications and technologies, including packaging [147,148,149,150,151], biomedical applications [82,152,153,154], printed electronics [155,156,157], and as an additive for paper coating technology [158,159,160]. However, because of the hydrophilicity of nanocellulose, its mechanical and barrier properties [161,162,163,164,165,166,167,168,169] are reduced when exposed to a humid environment [167]. Aside from chemical treatments [162,169,170,171,172], to resolve this problem and improve the barrier properties of CN films, the fabrication of hybrid nanocomposites using nanocellulose and nanoclay particles is an alternative to reduce the cost and enhance the physical and mechanical properties of the composite structures made from nanocellulose. In this hybridization process, the final properties of the hybrid nanocomposites made from nanocellulose and nanoclay are highly dependent on the quality of mixing and exfoliation of the nanoclay particles [173], which subsequently determine the final morphology.

Moreover, the interactions between the CN and the nanoclay particles must be well-understood [174]. Different types of nanoclay, including montmorillonite [174,175,176,177,178,179,180,181,182,183,184], saponite [180,185,186], bentonite [187], aminoclay [188], and vermiculite [189], have been used to fabricate hybrid nanocellulose-based nanocomposites. In some studies, the hybrid nanocomposites were made by adding another component, such as chitosan [177,190] or a polymer [191,192,193,194,195,196,197].

In terms of applications, various studies have attempted to enhance the barrier properties [175,176,177,178,181,182,183,184,186,188,189,190,192,193,195,196,197], mechanical strength [175,176,177,178,179,181,182,183,185,186,187,188,189,190,191,193,194], and fire retardancy [175,177,178,193,194] of nanocellulose-based hybrid composite films. Most of these studies are related to the montmorillonite nanoplatelets. There are similarities between building block materials, nanocellulose, and nanoclay for the fabrication of hybrid nanocomposites, and the differences mostly relate to the processing of the materials and, in some cases, chemical treatments of the initial materials and self-assembly of the components to enhance the physical and mechanical behavior of the hybrid nanocomposites. Here, the fabrication processes and material developments for hybrid nanocomposites based on nanocellulose and nanoclay are described.

Liu et al. developed a multi-layer clay nanopaper hybrid composite with gas barrier and fire retardancy properties by using a continuous cellulose nanofiber matrix filled with ordered montmorillonite platelets [175]. The continuous fibrillar matrix provided enhanced toughness, making it possible to increase the nanoclay content to very high concentrations while maintaining the tensile strength. The nanopaper was processed at room temperature using water as the processing medium. The presence of the clay nanoplatelets enhanced the oxygen barrier properties at higher humidity levels compared with the pure nanocellulose.

Transparent and flexible hybrid nanocomposite films with excellent mechanical behavior and gas barrier properties were developed by Wu et al. from montmorillonite nanoplatelets and 2,2,6,6-tetramethylpiperidine-1-oxyl radical (TEMPO)-oxidized cellulose nanofibrils using a mixing and drying process in which the nanoplatelets were dispersed in water using mechanical agitation without any surfactant [176]. A sample of the nanocomposite films containing 5% nanoplatelets exhibited a tensile strength of 509 MPa and Young’s modulus of 18 GPa. By taking advantage of ionic interactions between cationic nanocellulose and anionic nanoclay, Jin et al. developed a hybrid nanocomposite structure containing 63 wt. % nanoclay, resulting in a compressive strength of 76 MPa [179], which could be used in transportation [198] and construction. Using the assembly of montmorillonite nanoplatelets and nanocellulose fibrils from TEMPO-mediated oxidation, Wu et al. developed hybrid composite films that exhibited a larger water contact angle as compared with films made from pure components [180]. The large water contact angle on the surface of the nanocomposite films was explained using Cassie’s law, and based on the air area fractions of the surfaces of the composite films analyzed using atomic force microscopy. Yang et al. developed flexible and transparent hybrid layered nanocomposite films with high mechanical strength and barrier properties based on cellulose and montmorillonite nanoplatelets from an aqueous solution of cellulose/LiOH/urea [181]. A nanocomposite film containing 15% nanoplatelets showed the highest tensile strength. Interestingly, the hydrophilic nature of films made from pure cellulose was changed by incorporating the nanoplatelets. The composite films exhibited a hydrophobic behavior, which was attributed to the change in the orientation of cellulose chains on the surface of the composite films.

Using a simple and scalable method, low-cost, flexible, and mechanically strong hybrid nanocomposite films with low water vapor permeability were developed by Garusinghe et al. from nanocellulose and montmorillonite nanoplatelets [182]. The low vapor permeability was attributed to the increase in the tortuosity of the reinforcement structure that was achieved by increasing the quality of dispersion of nanoplatelets within the nanocellulose matrix. High-strength, sustainable hybrid nanocomposite membranes with good barrier properties for packaging applications were developed by Tayeb and Tajvidi using the evaporation-induced self-assembly of montmorillonite nanoplatelets and cross-linking reactions of two cross-linkers (Acrodur thermoset acrylic resin and polyamidoamine epichlorohydrin) at the interfaces of nanocellulose [183]. Shanmugam et al. developed a method for the rapid fabrication of hybrid composite sheets based on nanocellulose and montmorillonite nanoclay using a spray coating process [184]. This production method of hybrid nanocomposites is scalable and important for moving toward large-scale production of nanocomposite films.

Mechanically strong transparent ternary hybrid nanocomposite films with flame-retardant characteristics were developed by Liu and Berglund from high contents of montmorillonite nanoplatelets in a matrix of carboxymethyl cellulose sodium salt and a network of nanofibrillated cellulose in a water-based process [178]. Liu and Berglund used protonated chitosan in addition to nanocellulose and montmorillonite nanoplatelets [177] and found that, because of the ionic interactions between the components, adding chitosan to the mixture of clay and nanocellulose resulted in a rapid flocculation which reduced the filtration time.

In another study, by Enescu et al., evaporation induced self-assembly was used to fabricate ternary hybrid composite films with improved mechanical and barrier properties from CNCs, montmorillonite, and chitosan [190]. A limited number of polymers were used in ternary composites made from nanocellulose and nanoclay, which were poly(vinyl alcohol) [194,197], poly(lactic acid) [192,196], epoxy [193], and PP [195]. The objectives of these studies were to improve the barrier [192,193,195,196,197] and mechanical properties [191,193,194,195]. Liu et al. used aminoclay and carboxylated cellulose nanofibrils to fabricate strong, flexible, and transparent hybrid nanocomposites films [188]. These nanocomposite films exhibited excellent mechanical properties, which were caused by the ionic bonds formed between amine groups of clays and carboxylic groups of the nanocellulose. This shows the importance of the interaction between the nanoclay and nanocellulose, and it significantly effects the physical and mechanical behavior of hybrid nanocomposites.

### 3.4. Nanocellulose and Polyhedral Oligomeric Silsesquioxane Hybrids

Polyhedral oligomeric silsesquioxane (POSS) is a hybrid nanofiller whose structure contains both organic and inorganic groups with potential applications in making nanocomposites. Compared with other nanofillers, POSS can be more easily modified to create various types of structures [199]. Research related to the fabrication of hybrid nanocomposites based on POSS and nanocellulose is not as extensive as it is for other nanofillers such as nanoclay. Most of the related studies have used reactive or functionalized POSS as a nanofiller to enhance the properties of nanocellulose [200,201,202,203,204,205,206,207,208,209]. Depending on the functionality of POSS, there is a possibility of grafting and cross-linking reactions with the nanocellulose matrix that can significantly change the behavior of the resulting nanocomposite. Reactive POSS nanoparticles have been used to make cellulose-based hybrid composites, including POSS with multi-N-methylol groups [200,206,207,208,209,210], octa-amino-propyl silsesquioxane (POSS–NH_2_) [203,205], and POSS-modified cellulose acetate [201].

In terms of properties and applications, hybrid nanocomposites based on POSS and cellulose have been made to improve the thermal [203,204,205], adsorption [206,208,210], and shape memory [200] properties and for biomedical [211] and energy [212] applications. Most studies on hybrid nanocomposites based on nanocellulose and POSS were conducted by Xie et al. [200,205,206,207,208,209,210]. These studies include using reactive POSS of different kinds to make nanocellulose-based hybrid composites to improve the thermal [205,209], adsorption [206,208,210], and shape memory properties [200]. For example, Xie et al. fabricated hybrid nanocomposites made from reactive POSS and nanocellulose through grafting and cross-linking reactions where POSS–NH_2_ was used as the reactive POSS, and dimethylol dihydroxy ethylene urea was used as the cross-linking monomer [205]. Xie et al. fabricated hybrid nanocomposites with improved adsorption properties to remove reactive dyes [206] and C.I. Reactive Red 250 [210] from aqueous solutions, and to remove Cu^2+^ and Ni^2+^ from waste water [208]. Another class of hybrid nanocomposites based on POSS and cellulose is a transparent ternary hybrid nanocellulose-based nanocomposite film containing both montmorillonite and POSS with amino groups, which was developed by Shin et al. [213]. They found that the presence of both montmorillonite and POSS in the hybrid nanocomposite films reduced the transparency of films to UV rays from the sun, which makes this hybrid film a potential candidate for a sustainable material for UV protection.

### 3.5. Nanocellulose and SiO_2_ and Other Nanomaterials Hybrids

Researchers have used other nanomaterials for combinations with nanocellulose, including SiO_2_, antimony tin oxide, and silver nanoparticles. For example, Fang et al. [214] used nanocellulose and antimony tin oxide to reinforce polymer PVA. The obtained composites could be used as personal thermal management textiles since antimony tin oxide is an n-type semiconductor material. Compared with the tensile strength of neat PVA (~16 MPa), the tensile strength of the composite was much higher (~26 MPa) [214]. Table 3 shows the main preparation conditions and properties of the hybrid polymer composites.

Zhao et al. [217] used nanocellulose and nano-SiO_2_ to reinforce polymer polyacrylic acid (PAA). The nanocellulose inhibited aggregation of SiO_2_ in PAA. In addition, the cross-linking by hydrogen bonding improved the interfacial compatibility of the composite system. Figure 8 shows a proposed mechanism between the nanocellulose/SiO_2_ hybrid fiber and PAA matrix. Compared with the individual nanocellulose and SiO_2_, the nanocellulose/SiO_2_ hybrid improved the mechanical performance of the composites [217].

In another study, Zhang et al. [218] used nanocellulose and silver nanoparticles to reinforce poly(ethylene imine). Silver nanoparticles are a noble metal material that can catalyze the discoloration of organic dyes and are known to be an effective anti-microbial agent. When nanocellulose was combined with silver nanoparticles and poly(ethylene imine), strong hydrogen bonding and interactions between the amine, hydroxyl, carbonyl groups, and silver particles provided a stable 3D network (Figure 9). The obtained composites exhibited excellent shape retention and mechanical stability in wet conditions [218].

Cheng et al. [219] used nanocellulose and silver nanoparticles to reinforce polyurethane. Nanocellulose aided in the uniform dispersion of silver nanoparticles within the polyurethane matrix. The obtained composites could be used as antibacterial coatings in wood materials since silver nanoparticles have no toxicity and are widely used for antimicrobial materials [219]. Adel et al. [215] used nanocellulose, silver nanoparticles, and sodium alginate to reinforce a paper matrix. The nanocellulose/silver/sodium alginate dispersed homogeneously on the paper surface. The obtained composites exhibited excellent thermal, mechanical, antibacterial, and barrier properties. For example, after the coating of nanocellulose/silver/sodium alginate, the paper composites had a higher tensile index (by 16%) and Young’s modulus (by 11%) [215]. Wang et al. [216] used nanocellulose and silver nanoparticles to reinforce polymer PVA. The film composites exhibited a tensile strength of 2.4 MPa and strain of 242%. In addition, the film composites were biodegradable and exhibited strong antimicrobial properties [216].

## 4. Applications

This section presents specific applications for hybrids of nanocellulose with micron-sized fibers and nanoparticles in five application areas.

### 4.1. Mechanical Reinforcement Applications

The hybridization of reinforcing fibers with nanocellulose has been used to improve the mechanical properties of fibers. Eom et al. [220] used nanocellulose for the multiscale hybridization of natural silk to improve mechanical properties. The mechanical strength of the hybridized composite increased by 110%, and the impact strength increased by 228%. The silk fiber/CNF composite maintained its structural stability in a water environment and could be used as a reinforcement in plastic composites.

Jabbar et al. [221] prepared epoxy composites reinforced with hybridized jute fibers and a nanocellulose coating to improve their tensile modulus (21%), flexural strength (47%), flexural stiffness (48%), and fracture toughness (32%). However, the tensile strength of the composites was reduced compared with the uncoated jute fiber epoxy composite.

Fibers have also been hybridized with nanocellulose to facilitate better adhesion with the polymer matrix by improving surface morphology.

Bang et al. [78] produced bamboo fiber/cellulose nanofiber composites using thermocompression and used the composite to reinforce PBS polymer. The nano-brush morphology of the bamboo fiber/cellulose nanofiber reinforcement maximized the interaction with the polymer matrix and increased the bonding between the bamboo fibers without the use of a synthetic binder or cross-linker. The tensile strength of the composite with 15% reinforcement of bamboo fiber/cellulose nanofiber was increased by 240%, and Young’s modulus was increased by 700% compared with the neat polymer.

De Souza and Tarpani [222] spray coated cellulose nanofibrils onto GF and CF fabrics to create a hybridized hierarchical reinforcement system. The hybridized hierarchical fabrics were used to reinforce epoxy resin. The nanocellulose/GF-epoxy composite improved tensile strength and tensile strain at break, increasing the tensile strength by 75%. Similarly, the flexural strength was increased by 65%.

Pommet et al. [223] used *Acetobacter xylinum* bacteria to directly deposit bacterial nanocellulose (BNC) around sisal and hemp natural fibers. The resulting hierarchical fiber-reinforced nanocomposites had better surface adhesion to the cellulose acetate butyrate (CAB) and poly(L-lactic acid). The IFSS was determined by the fiber pull-out test of sisal-BNC with CAB and was 46% higher than the IFSS of sisal with CAB. Similarly, the IFSS of hemp-BNC with CAB was 141% higher than hemp with CAB. The IFSS of sisal-BNC with PLA was 21% higher than sisal with PLA.

Hybridization with nanocellulose has also been used to aid the manufacturing process. Park et al. [224] used hybridized nanocellulose-reinforced polyacrylonitrile (PAN) as a precursor to producing CFs. The hybridized nanocellulose/PAN precursor produced increased graphite crystalline size at the same carbonization temperature compared with the PAN-only precursor. The electrical conductivity and the mechanical properties of the resulting CFs were also improved. The incorporation of nanocellulose in PAN has the potential to decrease energy consumption in CF production by lowering the carbonization temperature.

### 4.2. Electronic Applications

Wang et al. [225] developed core-shell structured nanocellulose/CNT hybrids for use as flexible supercapacitors. The nanocellulose hybrid was used to create self-healable electro-conductive hydrogels with high mechanical toughness, excellent viscoelastic characteristics, and ideal electroconductivity. The hydrogels provide an alternative platform for personal wearable electronic devices.

Liu et al. [226] developed a high-yield process for the manufacturing of a high-conductive graphene/nanocellulose hybrid by co-exfoliation of low-oxidized expanded graphite and microfibrillated cellulose (LGENC). The hybrid reached a high in-plane conductivity of 5800 S/m but a very low out-of-plane conductivity. The anisotropic conductivity of LGENC creates possibilities for its application in the field of wearable, portable electronic equipment, and other electronic devices. LGENC may also be applicable as mechanical reinforcement for carboxylated nitrile butadiene rubber latex, as it increased the tensile strength of the latex by up to 324%, elongation at break by 34%, and fracture energy by 437%.

### 4.3. Packaging and Adhesive Applications

Errokh et al. [227] hybridized nanocellulose fibers with silver nanoparticles and used the hybridized nanocellulose–silver fibers to reinforce commercial acrylic latex for packaging applications. The hybridized nanocellulose–silver acrylic latex composite was effective in preventing bacterial growth on the surface of the composite while maintaining the stiffness required for packaging applications.

Mo et al. [228] used an Aramid/nanocellulose/softwood pulp hybrid to create composite insulating paper with low relative permittivity, reduced dielectric loss, good mechanical properties, and high thermal stability. Aramid pulp contributes to a reduction in relative permittivity and dielectric loss but also reduces mechanical properties. NCF acts as a mechanical reinforcement to compensate for the loss in mechanical properties by filling the void defects and improving the interfacial bonding. The hybrid composite paper had 30% lower relative permittivity and 43% lower dielectric loss compared with the conventional paper formed by pure softwood pulp. The hybrid composite paper also had better insulating and thermal properties compared with the conventional paper. The hybrid paper has the potential to replace conventional softwood paper in high-voltage alternate current (AC) transformer insulation.

Zakuwan and Ahmad [229] hybridized κ-carrageenan bioplastics with CNCs and organically modified montmorillonite to improve the mechanical properties of the bioplastics. The bionanocomposite had a 140% higher tensile strength than the neat biopolymer, and a 167% higher Young’s modulus than the neat polymer. The CNC/montmorillonite hybrid reinforcement reduced the water uptake of the biopolymer by 74%.

Ng et al. [230] used hybridized nanocellulose crystals derived from corncob with silver nanoparticles and nitrostatin to create poly(lactic acid)/nanocellulose nanocomposite films (PLA/CCNC/silver/NI). The films were very effective at inhibiting the microbial growth of *E. coli* and *S. aureus*. The silver ion migration was below the EFSA regulated level (<0.05 mg/L in food simulants). The nanocellulose hybrid films could have potential applications in food packaging.

Yu et al. [231] used cellulose nanofibril/carbon nanomaterial hybrid aerogels for the removal of cationic and anionic organic dyes as shown in Figure 10. Two classes of carbon nanomaterials, CNTs and GnPs, hybridized with cellulose nanofibrils, were efficient in removing cationic methylene blue and anionic Congo red dye in single and binary systems. The dyes could be easily desorbed from CNF-GnP aerogels using ethanol as the desorption agent, making the nanocellulose hybrid reusable. The CNF-GnP hybrid has potential use as an adsorption material for wastewater treatment.

### 4.4. Optical Applications

Schütz et al. [232] studied the formation of nanofibrillated cellulose and titania nanoparticle hybrids in an aqueous solution. The thin films formed by the nanoparticle hybrids had a very high Young’s modulus of up to 44 GPa and hardness of up to 3.4 GPa. The hybrid films had an optical transmittance of ~80%. These properties of the nanocellulose hybrids make them a potential candidate for applications that require high wear resistance and UV activity.

### 4.5. Medical Applications

Anton-Sales et al. [233] used BNC and titania hybrids (BNC/TiO_2_) to create cytocompatible cell carriers that can be cryopreserved. Viable cell cultures were obtained after thawing the BNC/TiO_2_ carriers stored in liquid nitrogen for more than a week.

Berndt et al. [234] produced antimicrobial porous hybrids with BNC and silver nanoparticles by firmly immobilizing silver nanoparticles on the top and bottom surfaces of the BNC film through chemical interactions. A medical dressing prepared from the BNC/silver nanoparticle hybrid showed strong antimicrobial activity against *E. coli*, and the activity was localized to the dressing, thus avoiding the release of the nanoparticles into the wound.

Zaid et al. [229] created a reduced graphene oxide/TEMPO-nanocellulose-based electrochemical biosensor for the determination of *Mycobacterium tuberculosis.* The nanohybrid films NH_2_-rGO/TEMPO-NCC, immobilized onto a screen-printed carbon electrode, distinguished between positive and negative samples of *M. tuberculosis.* The results showed that the nanocellulose hybrid sensor had high sensitivity and was extended to analyze real PCR (polymerize chain reaction) products.

## 5. Challenges and Opportunities

Since the discovery of nanocellulose in the 1980s, and especially in the early 2000s, numerous studies from around the world by various research groups have been published and are increasing exponentially. Nanocellulose has numerous applications, as described in the previous section. Some new commercial products use nanocellulose, including in automotive applications (e.g., Ford Motor Co., USA; Nanocellulose Vehicle, Japan), inks, diapers, and medical applications.

However, there are some challenges in the processing of nanocellulose and fine dispersion of nanocellulose in polymer matrices. The current pricing of nanocellulose is another barrier for the full commercialization of nanomaterials.

Moreover, concerns about the toxicology of nanomaterials and the safety of nanocellulose and nanocellulose composites are addressed in this section; no significant toxic effects have been observed.

### 5.1. Safety of Nanocellulose and Nanocellulose Composites

Bio-based CNs are sourced from natural materials, creating an opportunity for the use of more biodegradable, renewable, and sustainable materials in a wide range of applications. Generally, research on unmodified CNs has demonstrated that, similar to conventional celluloses, CNs are relatively inert and biocompatible. In composites, the toxicity of embedded CNs depends on the concentration and form upon release, potential for release, and environment into which the materials are released. However, given the low toxicity and limited potential for release of ultrafine particles that can be inhaled (<10 µm) or are respirable (<50 µm), the overall concern about health risks from inhaling CNs in composites is expected to be low under most conditions.

Safety of CNs: Most existing research on CN toxicology and exposure indicate that CNs are relatively safe and do not introduce novel hazards when released into the environment, ingested, or applied to the eyes or skin. Studies suggest a low environmental risk since cellulose nanofibers and CNCs are demonstrated to not have an adverse effect on various aquatic or microbial life, even at high concentrations [235,236,237,238]. No adverse effects were observed after consumption of CNs at up to 4% of the diet in rats [237,239,240], after dermal or eye exposure to CNs [237,239], or in genotoxicity studies [239,241,242]. Studies of conventional celluloses demonstrate that even long-term consumption of cellulose powder, microcrystalline cellulose powder, and microcrystalline cellulose gel consisting of 30% of the diet in rats for 72 weeks did not result in adverse effects. One of the main entry points into the human body is likely to be inhalation during occupational exposure [243,244]. There are some uncertainties related to the inhalation toxicity of CNs. Conventional cellulosic materials are poorly soluble, low-toxicity materials which are known to cause lung irritation when inhaled [245]. Similarly, CNs also irritate the lungs and can cause short-term and transient inflammatory effects [245,246]. Studies on the biopersistence of conventional cellulose and CNs indicate that cellulose may be biodurable and persist in the lungs [247,248], which can lead to chronic health issues with repeated and long-term exposure to dust. Inhalation exposure limits need to be established; however, there are challenges associated with dispersing and performing realistic inhalation experiments [247]. Some results demonstrate that there may be slight differences in uptake, toxicity, and accumulation related to different physicochemical characteristics, such as aspect ratio and changes to surface chemistry [236].

Safety of CN composites: When nanomaterials are incorporated into composites, individual nanoforms are generally not released; nanomaterials are usually released as aggregates, binding with other matrix materials [249]. The magnitude and characteristics of released particles depend on a variety of factors, including the composite matrix, content, dispersion, type, functionalization, challenge, and energy input, which vary based on the intended application of the product. For example, food packaging, coatings, cosmetics, and concrete each comprise different materials and undergo different types of use, wear, and means of disposal. Releases from high-energy mechanical manipulation of composites (e.g., sanding, drilling, cutting), aerosolization, and/or after exposure to high temperatures (e.g., tires, additive manufacturing) may result in exposure to finer composite particles.

Release scenarios may include intentional manipulation of composites (e.g., tearing, grinding), natural weathering, and use of composites. Few studies have measured the release potential of CNs from various composites. Studies demonstrate limited worker exposure to airborne nanoparticles during the manufacturing of CNs, including grinding and spray drying of microfibrillated cellulose [250,251]. Studies on other carbonaceous, metal-oxide, and organic nanomaterials have demonstrated a low release of nanomaterials that are mostly bound to other materials in aggregates [249,252]. In a quantitative evaluation of release rates from nanocomposites, the matrix material was the primary factor that determined rates of release, varying across five orders of magnitude. The nanomaterial additives had a much lower influence on particle release, varying release rates by a factor of less than 10 [249]. Interestingly, some high-aspect ratio materials (e.g., CNTs) reinforced some matrix materials (e.g., cement) and resulted in lower release rates of larger particles compared with the matrix material alone [249,253]. Similarly, fibrillar CNs can stabilize and reinforce composites because of their high surface area and physical entanglement with other fibers and materials [254,255,256], which may also result in lower release rates as compared with the matrix material. However, more research is required to understand the relationship between the integration of CNs in composites and particle release.

Human exposure to CNs is most likely to occur during the manufacture of raw CNs and CN composites, and during any use involving high-energy processes such as sawing, cutting, drilling, and sanding [244,252,257]. Slow release due to weathering and aging (e.g., from concrete) is not likely to result in significant exposure. During the manipulation of composites, it is recommended that any inhalable or respirable releases are controlled through the use of engineering controls. As with the handling of all products containing poorly-soluble fibers, respiratory protection is recommended during the sanding, cutting, drilling, and milling of CN composites in the presence of inhalable or respirable particulates (between 10 and 50 µm). Existing personal protection equipment, such as respirators, is typically used for handling conventional materials and has been demonstrated to be adequate for use with nanomaterials [258]. Typical control measures such as fume hoods and the proper use of equipment reduce exposure to airborne CN particles [250,259]. Currently, data on exposure or inhalation toxicity are insufficient to determine an occupational exposure limit for CNs [257].

One of the key challenges to performing exposure studies is the accurate and sensitive detection and quantification of carbon-based CNs, particularly unmodified CNFs. Although some methods and standards have been developed to measure other nanomaterials, the detection of CNs remains a challenge. Some approaches to detecting and labelling CNs have been developed, including the use of particle counters, electron microscopy, carbon analysis and gas chromatography with mass spectrometry, and fluorescence- and radio-labeling [251,259,260,261,262,263]. Improved detection of CNs will support more accurate calculation of their release from consumer products and in occupational settings.

### 5.2. Opportunities for Nanocellulose in Hybrid Composites

Composite materials provide many breakthrough applications, and hybrid fiber reinforcement provides many new solutions and concepts for engineering problems. As the most abundant material available in nature, nanocellulose can provide many benefits, which are further improved by the addition of commercial fibers and nanomaterials. With many new ideas and inventions by researchers around the world, more findings will result in more commercial applications.

The use of hybrid systems creates a synergistic effect in mechanical performance, thermal performance, and many other physical and chemical properties of the materials. Aerospace, automotive, medical, electronic, and defense are just a few of many possible application areas.

## 6. Concluding Remarks

Nanomaterials and nanocellulose have been one of the most important topics in the scientific community for more than 20 years, affecting many fundamental science and engineering disciplines. The number of nanocellulose publications has increased significantly over the years, as shown in Figure 11, and there are many new applications in construction, packaging, transportation, and electronics.

Some commercial products are already available in the market using nanocellulose (not a hybrid form) in paper coatings and cement formulations. Moreover, in the medical industry, various products use bacterial cellulose. Recently, the automotive industry has started using cellulose nanofibrils in polyurethane formulations, and the final products are commercially available.

The hybridization of cellulose nanofibers provides opportunities for new applications with improved properties and synergistic effects in various uses. The interaction of nanocellulose with commercial fibers provides improved mechanical properties, thermal properties, acoustic properties, and others. The synergistic use of other nanomaterials with CNs provides many new advanced materials and applications, such as electronic uses, in batteries, in sensors, and in environmental protection.

In this review, the hybrid approach of using nanocellulose and other materials is documented. These composites have applications in various fields, demonstrating the importance of these important sustainable materials.

## Figures and Tables

**Figure 1 polymers-15-00984-f001:**
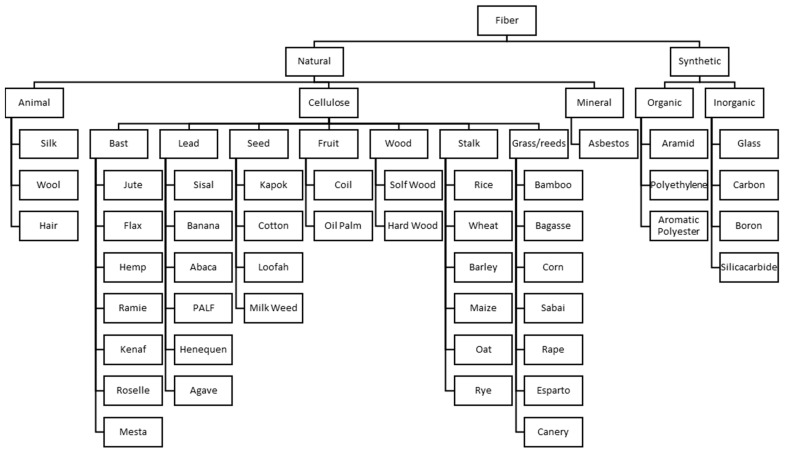
Classification of various fiber types [23].

**Figure 2 polymers-15-00984-f002:**
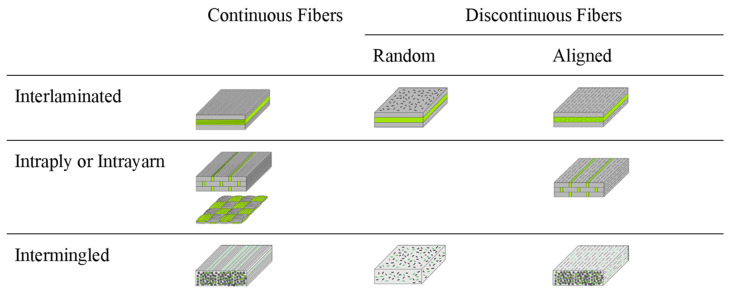
Hybrid fiber structures [28].

**Figure 3 polymers-15-00984-f003:**
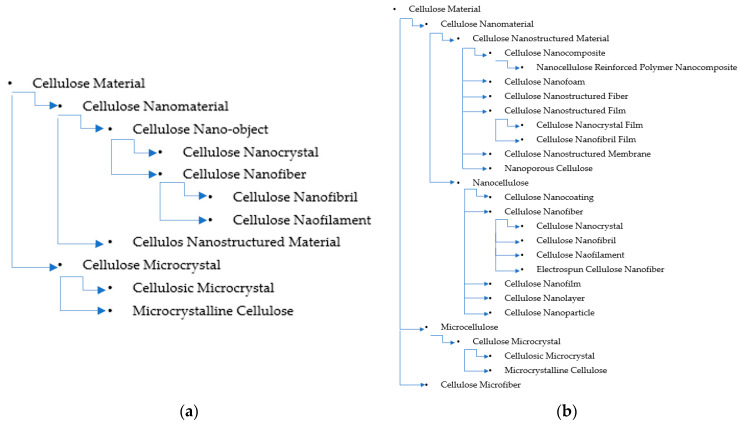
Cellulose material taxonomies (**a**) classification by ISO/TS 20477-2017 and (**b**) modified classification used in this review.

**Figure 4 polymers-15-00984-f004:**
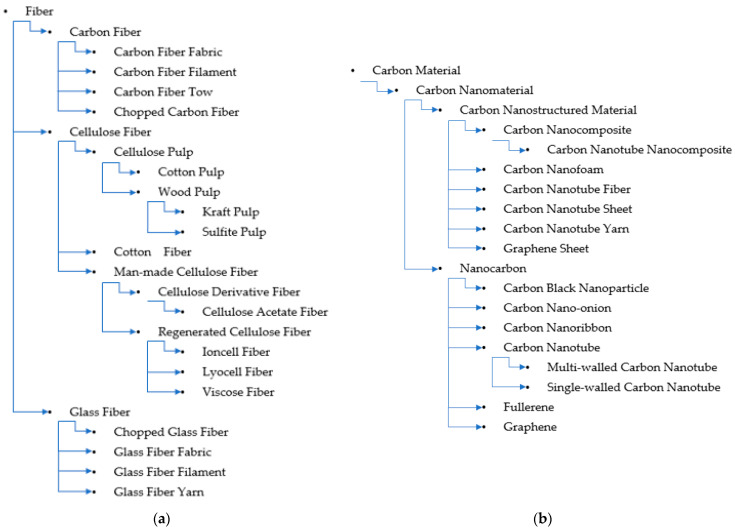
(**a**) Fiber taxonomies and (**b**) carbon nanomaterial classification used in this review.

**Figure 5 polymers-15-00984-f005:**
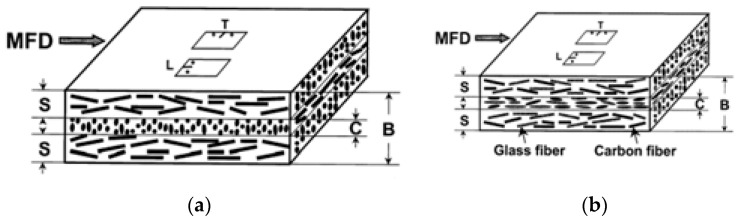
Schematic illustration showing the three-layer structures for (**a**) single-additive and (**b**) hybrid composites with GFs/CFs [57].

**Figure 6 polymers-15-00984-f006:**
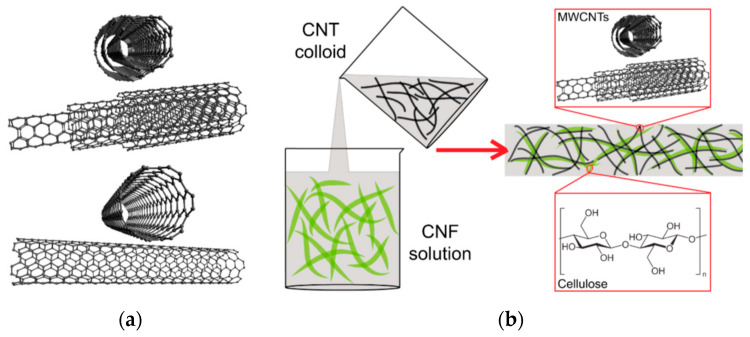
Scheme of (**a**) multi-walled and single-walled CNTs and (**b**) the preparation and structure of nanocellulose/CNT composites [111].

**Figure 7 polymers-15-00984-f007:**
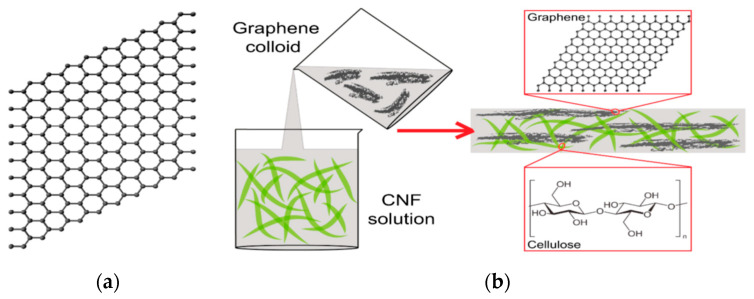
Scheme of (**a**) graphene and (**b**) the preparation and structure of nanocellulose/graphene composites [111].

**Figure 8 polymers-15-00984-f008:**
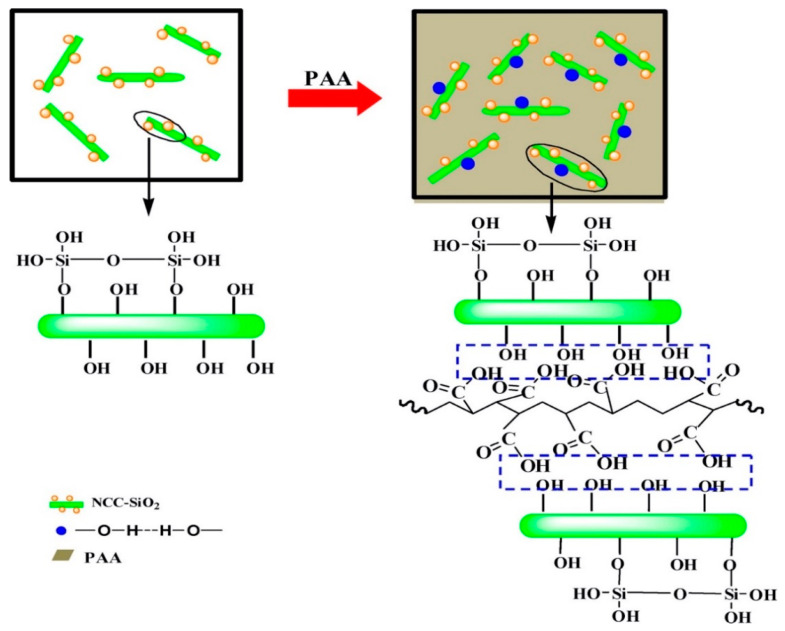
A proposed mechanism between the nanocellulose/SiO_2_ hybrid fiber and PAA matrix [217].

**Figure 9 polymers-15-00984-f009:**
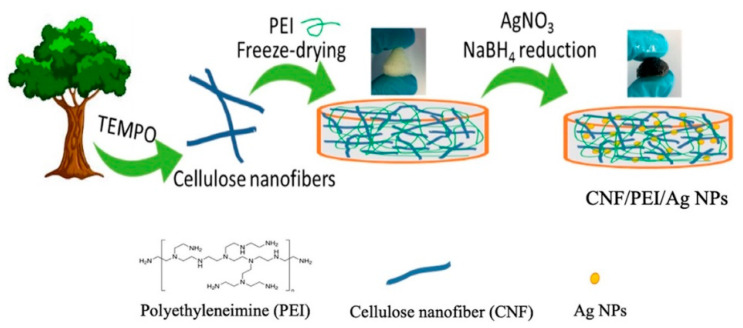
A scheme of the composite fabrication [218].

**Figure 10 polymers-15-00984-f010:**
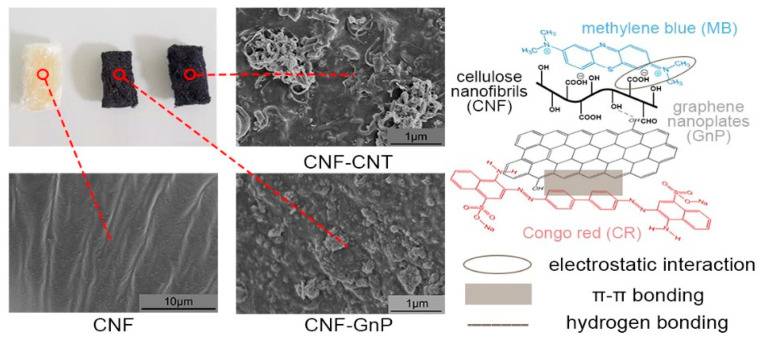
Mechanism of CNF-GNP hybrid aerogels for removal of cationic and anionic dyes [230].

**Figure 11 polymers-15-00984-f011:**
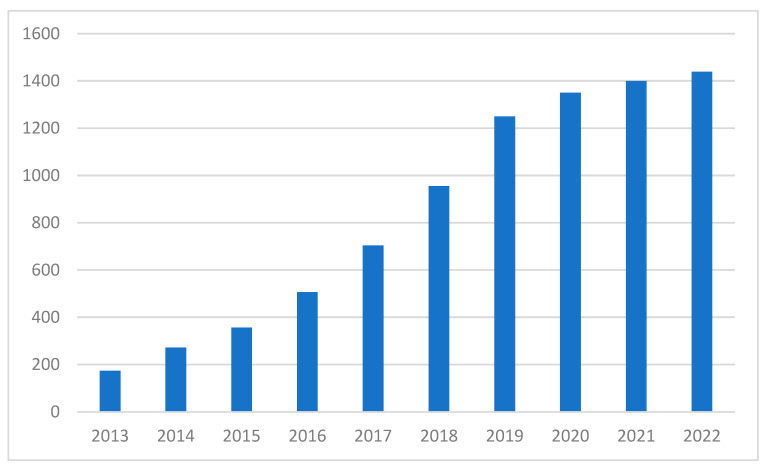
Number of nanocellulose publications with respect to years [264].

**Table 1 polymers-15-00984-t001:** Mechanical properties of hybrid composites [25].

Hybrid Fiber	Fiber Length (mm)	Fiber Content (wt %)	Resin	Chemical Treatment	Tensile Strength (MPa)	Tensile Modulus (GPa)	Flexural Strength (MPa)	Flexural Modulus (GPa)
PALF*-glass	—	25	Polyester	Untreated	72	—	101.2	—
Sisal-glass	—	30	Polyester	Alkali	130	—	150.0	—
Sisal-silk	10	—	Unsaturated Polyester	Untreated	17	—	33.5	—
Sisal-silk	10	—	Unsaturated Polyester	Alkali	21	—	50.5	—
Jute-glass	—	14	Unsaturated Polyester	Untreated	—	—	—	—
Bamboo-glass	—	30	Polypropylene	Untreated	18	3	34.0	3.4
Bamboo-glass	—	30	Polypropylene	MAPP*	19	3	42.0	4.5
Palmyra-glass	30	41	Rooflite	Untreated	26	1.4	44.5	1.38
Palmyra-glass	40	32	Rooflite	Untreated	26	1.4	26.7	1.55
Coir-glass	20	—	Phenolic	Untreated	22	3.8	53.4	4.8
Coir-glass	20	—	Phenolic	Alkali	26	4.3	68.3	5.6
Sisal-glass	—	Silk fiber: 20 Glass fiber: 10	Polypropylene	Untreated	30	2.3	66.7	4.03
Roystonea regia Glass	7	Roystonea regia fiber: 15 Glass fiber: 5	Epoxy	Untreated	32	2.4	40.1	3.9
Glass-sisal	30	—	Polyester	Untreated	176	—	—	—
Sisal-glass	35	—	Epoxy	Untreated	69	—	—	—

PALF*: Pineapple Leaf Fiber; MAPP*: Maleic-Anhydride-Polypropylene.

**Table 2 polymers-15-00984-t002:** Processing methods for hybrid composites.

Thermoset Processing	Thermoplastic Processing
Autoclave molding	Extrusion
Cold pressing	Injection molding
Compression molding	Thermoforming
Hand-layup	Additive manufacturing
Vacuum bagging	Prepreg
Vacuum assisted resin transfer molding	
Spray-up	
Filament winding	
Prepreg	
Pultrusion	

**Table 3 polymers-15-00984-t003:** The main preparation conditions and properties of the hybrid polymer composites.

Hybrid Fiber	Polymer	Main Preparation Conditions	Main Property of Composite	Ref.
Nanocellulose/antimony tin oxide	PVA	Blend solutions of fibers and polymer → ultrasonic mixing → dry at room temperature	Tensile strength: ~26 MPa Strain: 16%	[214]
Nanocellulose/silver/sodium alginate	Cellulose	Mix nanocellulose suspension with silver nitrate solution → add sodium alginate solution → coat on paper sheet → dry	Tensile index: 795 N·m/g Young’s modulus: 3038 MPa	[215]
Nanocellulose/silver	PVA	Mix nanocellulose slurry with silver nitrate solution → mix with PVA solution → cast → dry	Tensile strength: 2.4 MPa Strain: 242%	[216]

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
