# Peer review of "Review on Hybrid Reinforced Polymer Matrix Composites with Nanocellulose, Nanomaterials, and Other Fibers"

_polymers, 2023, doi:10.3390/polym15040984_

Round 1
Reviewer 1 Report
This research reports on a review on hybrid reinforced polymer matrix composites containing various constituent materials and their mechanical characterization. This is an interesting topic which provides useful information on FRP composite materials and structures. However, there are important issues that need to be addressed by the authors through Major Revision in the interest of improving this work before I can recommend this manuscript for publication. My comments/recommendations are as follows:1) The authors have talked about the optimization process of FRP composites several times, but the reviewer can't see any literature done on this topic. 2) Please discuss how various fiber types can influence the mechanical properties and response of FRP composites. 3) The reviewer believes that the section associated with the manufacturing technique of FRP composite materials doesn't provide useful information into past and present manufacturing techniques of composite materials. There are various types of composite manufacturing processes including: open molding, closed molding, filament winding, Vacuum Infusion Process (VIP), hand layup, and cast polymer molding. There are a variety of processing methods within these molding categories, each with its own benefits.
4) The application of composite materials must be categorized based on the correlation between the types of composites, their manufacturability, and their industrial/engineering applications. 5) The reviewer believes the following references add value to the present study and ,thereby, must be discussed in the introduction portion of the manuscript: "Mechanical characterization of particulated FRP composite pipes: A comprehensive experimental study", Polymer Testing, Volume 93, January 2021, 107001.
"A semi-empirical approach to evaluate the effect of constituent materials on mechanical strengths of GFRP mortar pipes", Structures, Vol:36, Feb 2022, Pages 493-510.
"Optimization of the Mechanical Properties and the Cytocompatibility for the PMMA Nanocomposites Reinforced with the Hydroxyapatite Nanofibers and the Magnesium Phosphate Nanosheets", Materials2021, 14(19), 5893. 6) Conclusion doesn't support the state of art of this research. Please point out the novelty of this research article in this section. I believe responding to the above-mentioned comments and revising the manuscript accordingly are essential before considering the paper for publication.
Author Response
We are very much glad to receive all these important comments and suggestions. We are thankful to your review for for your great time and efforts. We have made significant changes to the manuscript. We have highlighted the revised part in the manuscript.
The questions are answered for each comment with respect to question numbers.
We are grateful for all the reviewers.
On Behalf of Team
Dr. M.Ozgur Seydibeyoglu
1) In the relevant section, the explanations are detailed.
2) This topic has been expanded by adding references.
3) There are numerous methods as you have explained. Manufacturing of composites is a very dynamic area and there are many new techniques discovered everyday (many of them are not even published). That would make the review much longer. We wanted to focus on nanocellulose hybrid composites. We mentioned an introductory part for the manufacturing.
4) Thank you for this valuable suggestion but at this stage it is not easy to reorganize the paper. It would require a major reorganization of the paper which is written by 14 authors. We have categorized for certain uses but the reviewer is absolutely right that it would be much better to correlate with types of composites, manufacturability, and other issues.
5) Thank you for the suggestions but these publications are not very closely related to natural fiber reinforced composites. Additional articles that can contribute are cited in the introduction part.
6) As this is a review article, we have summarized the recent trends in the conclusion with a graph of publications in the recent years. We have provided some commercial applications of nanocellulose and also we have demonstrated that the hybrid forms could have many new commercial uses providing many new opportunities for this material class.
Reviewer 2 Report
This manuscript by Seydibeyoglu et al. provides an extensive review of hybrid reinforced composites based on nanocellulose and other materials. In my opinion, it is well written and provides a useful contribution to the field, although there are a few issues that should be addressed before it can be published:
P1, L30: The authors state that 'composite materials date back to approximately 8,300 BC...'. I suggest that may be true for man-made composites, but many natural materials (e.g. wood, bone and shells) are also composites and I suspect their use dates back much further.
P2, L47-48: The authors state that 'natural fibers are primarily comprised of cellulose, hemicellulose, and lignin'. That is true of plant fibres, but some natural fibres (wool, hair, silk) are made of protein.
P2, L50-52: The reference to cellulose, hemicellulose and lignin repeats (almost verbatim) what was written a few lines earlier.
P5, L127-128: The authors state that 'natural fibers are hydrophilic because they contain hydroxyl groups on the surface and in the bulk'. That is true of cellulose, but not of the protein fibres, which are also hydrophilic to some extent.
P6, L169-174: In the section headed 'Manufacturing Methods', the authors provide a list of methods for forming composites, but do not convey the main challenges to be overcome.
I suggest that the essential step in forming a man-made composite is mixing the matrix materials with the fillers. It may be difficult to mix a filler into a high viscosity polymeric matrix; therefore low molecular precursors for the matrix are frequently used, which are polymerised after mixing the filler. Additionally, the composite must be shaped for its intended purpose. Short-fibre thermoplastic composites may be extruded or moulded; but this is not possible with long- or continuous-fibre composites, where some form of 'pre-preg' lay-up may be required.
In my opinion, it would be useful for the authors to expand this section to summarise the main difficulties to be addressed in composite manufacture.
P6-7, L179-180: 'Shanker and Rhim reported that nanofillers that can be used in the formation of hybrid nanobiocomposites.' This is not a complete sentence. (The simplest fix would be to remove one 'that'.)
Table 3: I am unsure why cellulose nanocrystals are given as amorphous - although their XRD patterns may be broad due to the small crystal sizes. Also, lignin is a semi-aromatic polymer, so I cannot see how oxidised lignin could be regarded as cellulose. Please explain or correct these entries.
P8, L220-221: The statement that 'cellulosic materials predominantly comprise cellulose' appears self-evident. I suggest the authors should clarify what they mean.
P14, L407: Grammatical or typographical slip: '...all of which are improve the IFSS compared...'.
P15, L451: 'PF' was previously designated in this review as 'plant fibres'. In what way are plant fibres synthetic? (Do the authors mean 'polymeric fibres' here?)
P18, L525-526: Grammatical or typographical error: There appears to be a conflict between singular and plurals in the phrase '...to create an environmentally friendly, low-cost, durable, dimensionally stable, non-melting, and non-toxic materials.'
P19, L559: The authors state (erroneously) that 'carbon is the most common element in nature'. Throughout the universe, hydrogen appears to be the most abundant element. From a terrestrial perspective, our planet contains iron (32.1% by weight), oxygen (30.1%), silicon (15.1%), magnesium (13.9%), sulfur (2.9%), nickel (1.8%), calcium (1.5%), and aluminium (1.4%); with the remaining 1.2% consisting of other elements. Even in dry plant tissue, oxygen constitutes around 45 %, while carbon makes up 44 %; and the proportion of oxygen will be higher in living plant tissue, due to the presence of water. The authors should clarify exactly what they mean, please.
Fig. 14: Since this review cannot be published before sometime in 2023, is it possible to update this graph to include the number of 2022 publications, please? (For example, could the authors count the publications using Web of Science, or a similar database?)
I will be happy to recommend publication, subject to addressing these few minor comments.
Author Response
We are very much glad to receive all these important comments and suggestions. We are thankful to you for your great time and efforts. We have made significant changes to the manuscript. We have highlighted the revised part in the manuscript.
The questions are answered for each comment.
We are grateful for your important questions and comments.
On Behalf of Team
Dr. M.Ozgur Seydibeyoglu
P1, L30: The authors state that 'composite materials date back to approximately 8,300 BC...'. I suggest that may be true for man-made composites, but many natural materials (e.g. wood, bone and shells) are also composites and I suspect their use dates back much further.
Sentence revised.
P2, L47-48: The authors state that 'natural fibers are primarily comprised of cellulose, hemicellulose, and lignin'. That is true of plant fibres, but some natural fibres (wool, hair, silk) are made of protein.
Sentence added.
P2, L50-52: The reference to cellulose, hemicellulose and lignin repeats (almost verbatim) what was written a few lines earlier.
Sentence deleted.
P5, L127-128: The authors state that 'natural fibers are hydrophilic because they contain hydroxyl groups on the surface and in the bulk'. That is true of cellulose, but not of the protein fibres, which are also hydrophilic to some extent.
Sentence added
P6, L169-174: In the section headed 'Manufacturing Methods', the authors provide a list of methods for forming composites, but do not convey the main challenges to be overcome.
I suggest that the essential step in forming a man-made composite is mixing the matrix materials with the fillers. It may be difficult to mix a filler into a high-viscosity polymeric matrix; therefore low molecular precursors for the matrix are frequently used, which are polymerised after mixing the filler. Additionally, the composite must be shaped for its intended purpose. Short-fibre thermoplastic composites may be extruded or moulded; but this is not possible with long- or continuous-fiber composites, where some form of 'pre-preg' lay-up may be required.
In my opinion, it would be useful for the authors to expand this section to summarise the main difficulties to be addressed in composite manufacture.
This is the introductory part for the review. We focus on hybrid composites with nanocellulose. If we focus on more manufacturing, the manuscript would be much longer.
P6-7, L179-180: 'Shanker and Rhim reported that nanofillers that can be used in the formation of hybrid nanobiocomposites.' This is not a complete sentence. (The simplest fix would be to remove one 'that'.)
That is deleted.
Table 3: I am unsure why cellulose nanocrystals are given as amorphous - although their XRD patterns may be broad due to the small crystal sizes. Also, lignin is a semi-aromatic polymer, so I cannot see how oxidised lignin could be regarded as cellulose. Please explain or correct these entries.
Removed Table 3
P8, L220-221: The statement that 'cellulosic materials predominantly comprise cellulose' appears self-evident. I suggest the authors should clarify what they mean.
Sentence deleted.
P14, L407: Grammatical or typographical slip: '...all of which are improve the IFSS compared...'.
Sentence revised.
P15, L451: 'PF' was previously designated in this review as 'plant fibres'. In what way are plant fibres synthetic? (Do the authors mean 'polymeric fibres' here?)
Abbreviation revised.
P18, L525-526: Grammatical or typographical error: There appears to be a conflict between singular and plurals in the phrase '...to create an environmentally friendly, low-cost, durable, dimensionally stable, non-melting, and non-toxic materials.'
Sentence revised.
P19, L559: The authors state (erroneously) that 'carbon is the most common element in nature'. Throughout the universe, hydrogen appears to be the most abundant element. From a terrestrial perspective, our planet contains iron (32.1% by weight), oxygen (30.1%), silicon (15.1%), magnesium (13.9%), sulfur (2.9%), nickel (1.8%), calcium (1.5%), and aluminium (1.4%); with the remaining 1.2% consisting of other elements. Even in dry plant tissue, oxygen constitutes around 45 %, while carbon makes up 44 %; and the proportion of oxygen will be higher in living plant tissue, due to the presence of water. The authors should clarify exactly what they mean, please.
Sentence revised.
Fig. 14: Since this review cannot be published before sometime in 2023, is it possible to update this graph to include the number of 2022 publications, please? (For example, could the authors count the publications using Web of Science, or a similar database?)
We have updated the graph with 2022 data. Thank you for this warning.
Reviewer 3 Report
This review paper is a very comprehensive introduction to the subject of Hybrid Reinforced Polymer Matrix Composites with Nanocellulose, Nanomaterials, and Other Fibers. The authors have made a really good job and I really enjoy the reading of this review paper. Although some revisions must be done before its acceptance for publication.
1. Introduction line 33 and line 38: merge the references
2. enlarge the analysis of Figure 1, Figure 3 and Figure 4
3. line 181 use the correct abbreviation for Carbon Nano Fibers (CNFs) not CaNFs
4. line 192 use - or / not both
5. Table 3 must be in one page and do not merge between to pages
6. line 272 use space before oC
7. Applications section and subsections: Add some figures in each subcategories of applications to help readers to understand better the mechanical applications, the electronic application, the packaging and adhesive applications and the optical applications.
8. Figure 14: I will prefer to saw this figure and the discussion in the biggening of nanocellulose section and not at the end
Best wishes for a happy new year
Author Response
We are very much glad to receive all these important comments and suggestions. We are thankful to you for your great time and efforts. We have made significant changes to the manuscript. We have highlighted the revised part in the manuscript.
The questions are answered for each comment.
We are grateful for your kind review.
On Behalf of Team
Dr. M.Ozgur Seydibeyoglu
This review paper is a very comprehensive introduction to the subject of Hybrid Reinforced Polymer Matrix Composites with Nanocellulose, Nanomaterials, and Other Fibers. The authors have made a really good job and I really enjoy the reading of this review paper. Although some revisions must be done before its acceptance for publication.
- Introduction line 33 and line 38: merge the references
Corrected
2. enlarge the analysis of Figure 1, Figure 3 and Figure 4
Revised the figures.
3. Line 181 use the correct abbreviation for Carbon Nano Fibers (CNFs) not CaNFs
Corrected.
4. line 192 use - or / not both
Corrected.
5. Table 3 must be in one page and do not merge between to pages
It is removed.
6. line 272 use space before oC
Corrected.
- Applications section and subsections: Add some figures in each subcategories of applications to help readers to understand better the mechanical applications, the electronic application, the packaging and adhesive applications and the optical applications.
It would be nice to add more pictures but the review was getting reall long so we had to make it in a certian limit. This is already almost 50 pages.
- Figure 14: I will prefer to saw this figure and the discussion in the beginning of nanocellulose section and not at the end
This is a conclusive graph about the status and the state of art for this material. It could be also nice to present it at the beginning but this is the authors’ choice and we believe it is also nice to present at the end as well.
Round 2
Reviewer 1 Report
The authors have done a great job of responding to my comments and revising the manuscript. As such, in my opinion, It is acceptable for publication in the present form.